# Targeting plasma membrane phosphatidylserine content to inhibit oncogenic KRAS function

Walaa E Kattan[1,3], Wei Chen[1], Xiaoping Ma[1], Tien Hung Lan[1], Dharini van der Hoeven[2], Ransome van der Hoeven[2], John F Hancock[1,3]

**The small GTPase KRAS, which is frequently mutated in human cancers, must be localized to the plasma membrane (PM) for biological activity. We recently showed that the KRAS C-terminal membrane anchor exhibits exquisite lipid-binding specificity for select species of phosphatidylserine (PtdSer). We, therefore, investigated whether reducing PM PtdSer content is sufficient to abrogate KRAS oncogenesis. Oxysterol-related binding proteins ORP5 and ORP8 exchange PtdSer synthesized in the ER for phosphatidyl-4-phosphate synthesized in the PM. We show that depletion of ORP5 or ORP8 reduced PM PtdSer levels, resulting in extensive mislocalization of KRAS from the PM. Concordantly, ORP5 or ORP8 depletion significantly reduced proliferation and anchorage-independent growth of multiple KRAS-dependent cancer cell lines, and attenuated KRAS signaling in vivo. Similarly, functionally inhibiting ORP5 and ORP8 by inhibiting PI4KIIIα-mediated synthesis of phosphatidyl-4-phosphate at the PM selectively inhibited the growth of KRAS-dependent cancer cell lines over normal cells. Inhibiting KRAS function through regulating PM lipid PtdSer content may represent a viable strategy for KRAS-driven cancers.**

## Introduction

RAS proteins are membrane-localized GTPases that regulate cell proliferation, differentiation, and apoptosis. RAS is a molecular switch that oscillates between an active GTP-bound and inactive GDP-bound state and functions as a critical node in growth factor receptor signaling pathways. Two classes of proteins regulate RAS.GTP levels: guanine nucleotide exchange factors activate RAS by promoting exchange of GDP for GTP, and GTPase-activating proteins stimulate RAS GTPase activity to return RAS.GTP to the inactive ground state. This regulatory circuit is subverted in 15–20% of all human tumors that express oncogenic RAS with mutations at

residues 12, 13, or 61 (Cox et al, 2014). These mutations block the ability of RASGAPs to stimulate GTP hydrolysis, thus oncogenic RAS is constitutively GTP-bound. HRAS, NRAS, KRAS4A, and KRAS4B (hereafter referred to as KRAS) are ubiquitously expressed in mammalian cells. These RAS isoforms have a near identical G-domain that binds guanine nucleotides and interacts with effector proteins, GTPase activating proteins, and guanine nucleotide exchange factors but have different C termini and membrane anchors. All RAS isoforms share a common in vitro biochemistry but exhibit different signaling outputs in vivo (Hancock, 2003). Reflecting these differences, each RAS isoform is mutated with different frequencies in different tumors. The major clinical problem is KRAS, which is mutated in >90% of pancreatic cancers, ~50% of colon cancers, and ~25% of non–small cell lung cancer (Prior et al, 2012).

To generate an output signal, RAS.GTP must recruit effector proteins from the cytosol to the plasma membrane (PM) for activation. One example is the MAPK cascade, where RAS.GTP recruits RAF to the PM for activation, in turn triggering the activation of MEK and ERK. Therefore, RAS proteins must be localized to the PM and correctly arrayed into nanoclusters for biological activity. Nanoclusters are transient RAS-lipid assemblies containing 5–6 RAS proteins that are the sites of effector activation (Murakoshi et al, 2004; Hancock & Parton, 2005; Plowman et al, 2005; Tian et al, 2007; Zhou & Hancock, 2015). KRAS is targeted to the PM by a C-terminal membrane anchor that comprises a farnesyl-cysteine-methyl-ester and a polybasic domain (PBD) of six contiguous lysine residues (Hancock et al, 1990). We recently used quantitative spatial imaging analyses and atomistic molecular dynamics to systematically examine the mechanism of association of this KRAS bi-partite PBD-prenyl membrane anchor with the PM. Traditionally, PBDs have been thought to interact with PM exclusively via electrostatics where the total number of basic residues determines the strength of electrostatic association with anionic lipids. However, we discovered that the molecular mechanism of KRAS PM binding is considerably more complex. The precise PBD amino acid sequence and prenyl group define a cryptic combinatorial code for lipid

[1]Department of Integrative Biology and Pharmacology, McGovern Medical School, University of Texas Health Science Center, Houston, TX, USA  [2]Department of Diagnostic and Biomedical Sciences, School of Dentistry, University of Texas Health Science Center, Houston, TX, USA  [3]The University of Texas MD Anderson Cancer Center UTHealth Graduate School of Biomedical Sciences, Houston, TX, USA

Correspondence: john.f.hancock@uth.tmc.edu

binding that extends beyond simple electrostatics; within this code, lysine and arginine residues are nonequivalent and prenyl chain length modifies nascent PBD lipid preferences. The code is realized by dynamic tertiary structures on the PM that govern amino acid side chain–lipid interactions and, thus, endow exquisite binding specificity for defined anionic phospholipids (Zhou et al, 2018). An important consequence is the ability of such anchors to sort or retain specific subsets of phospholipids into nanoclusters with a defined lipid composition. In this context, the structure of the KRAS anchor encodes exquisite binding specificity for phosphatidylserine (PtdSer) lipids with one saturated and one desaturated acyl chain (Zhou et al, 2014, 2017; Zhou & Hancock, 2015). The structure of the KRAS anchor, therefore, renders KRAS PM binding and, hence, KRAS function critically dependent on PM PtdSer content.

Preventing KRAS PM localization has been long advocated as an approach to block oncogenic function. However, early attempts to use farnesyltransferase inhibitors to prevent the first step of posttranslational processing that adds the KRAS membrane anchor failed because KRAS can be alternatively prenylated by geranylgeranyltransferase1 (GGTase1) when cells are treated with farnesyltransferase inhibitors (Hancock, 2003; Sebti & Der, 2003; Rowinsky, 2006). To identify an alternative strategy, we focused on the dependence of KRAS on PM PtdSer. We showed previously that indirect approaches to reduce PM PtdSer by manipulating sphingolipid and ceramide metabolism was moderately successful in reducing KRAS oncogenesis (Cho, 2016; van der Hoeven et al, 2017). Here, we evaluate direct targeting of the cellular machinery that actively maintains PM PtdSer content. PtdSer is the major anionic lipid on the inner leaflet of the PM comprising ~20 mol% of total lipid content (Vance & Steenbergen, 2005). The homologs oxysterol-related binding proteins, ORP5 and ORP8, encoded by *OSBPL5* and *OSBPL8*, respectively, are lipid transport proteins that function at membrane contact sites (MCSs) between the ER, PM, and other organelles. Both proteins transport PtdSer from its site of synthesis in the ER to the PM, where it is exchanged for phosphatidylinositol-4-phosphate (PI4P) (Filseck et al, 2015; Sohn et al, 2016). Therefore, we hypothesized that targeting PtdSer transporters ORP5 and ORP8 would disrupt KRAS PM localization and nanoclustering and attenuate KRAS function.

## Results

### Knockdown of ORP5 and ORP8 expression inhibits oncogenic KRAS signaling in vivo

We have shown previously that KRAS PM localization and nanoclustering are critically dependent on the PtdSer content of the inner PM leaflet (Zhou et al, 2014). ORP5 and ORP8 are lipid exchangers involved in the transport of PtdSer from the ER to the PM (Fig 1A); therefore, we hypothesized that inhibition of either protein will deplete the PM of PtdSer and inhibit KRAS signaling through mislocalization of KRAS from the PM. We first investigated whether PtdSer ER to PM transport is relevant for KRAS function in the model organism *Caenorhabditis elegans*, which expresses a single RAS

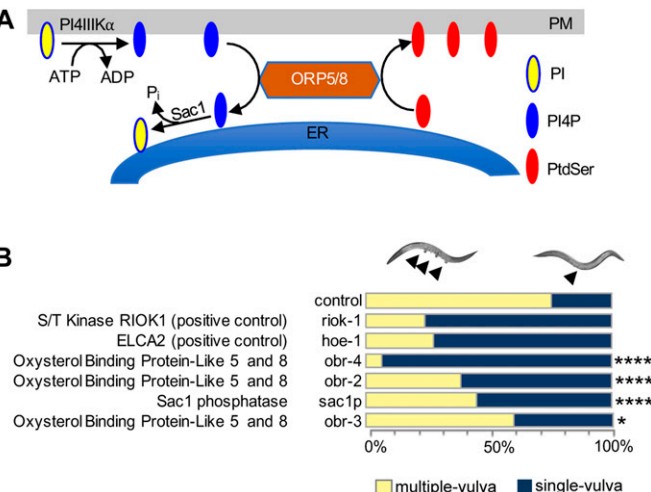

**Figure 1. ORP5 and ORP8 transport PtdSer to the PM.**
**(A)** *OSBPL5* and *OSBPL8* are paralogs that encode ORP5 and ORP8. These proteins are lipid transporters involved in lipid counter transport between the ER and the PM; they specifically exchange PtdSer in the ER with PI4P in the PM. The driving force of this process is a PI4P concentration gradient, whereby PI4P levels are high in the PM and are kept low at the ER by the action of the SAC1 phosphatase which immediately hydrolyzes PI4P. **(B)** RNAi knockdown screen of *OSBPL5*, *OSBPL8*, and *SAC1P* orthologs in an activated *let-60 C. elegans*. RNAi was induced by feeding let-60(n1046) L1 worms through adult stage with *E. coli* strain HT115, producing dsRNA to target genes. The presence of the MUV phenotype was scored using Differential Interface Contrast (DIC)/Nomarski microscopy. Previous reports show that h*eo-1* and *riok-1* potently suppress the *let-60* G13D MUV phenotype and, hence, were used as positive controls (****P < 0.0001, *P < 0.05). *OSBPL*, oxysterol-binding protein like; *obr*, oxysterol-binding protein related; SAC1P, SAC1-like phosphatidylinositide phosphatase.

gene, *let-60*, an ortholog of KRAS4B. We performed RNAi-mediated knockdown of validated orthologs of ORP5 and ORP8 in *C. elegans* carrying a mutationally activated G12D *let-60* allele (n1046), whose signaling leads to a multi-vulva (MUV) phenotype that is readily quantifiable. Because there are no clear homologs or orthologs of ORP5 and ORP8 in the worm, we performed blast analysis using the WormBase tool of the National Center for Biotechnology Information (NCBI) *OSBPL5* and *OSBPL8* sequences and obtained three hits: *obr-2*, *obr-3*, and *obr-4*. All three candidates affected the MUV phenotype when tested using RNAi with *obr-4*, demonstrating the strongest phenotype followed by *obr-2* and *obr-3* (Table S1). Knockdown of either *obr-4* or *obr-2* expression potently suppressed the MUV phenotype, with 93% and 62% of the population, respectively, displaying a single vulva. This extent of suppression is similar to the positive controls *riok-1* and *hoe-1*, previously described as potent suppressors of the MUV phenotype (Smith & Levitan, 2004; Weinberg et al, 2014) (Fig 1B). The enzyme Sac1 phosphatase resides in the ER and hydrolyzes PI4P to PI, creating a PI4P concentration gradient where it is high in the PM and low at the ER. This concentration gradient is also the driving force of ORP5/8 function (Filseck et al, 2015). Concordantly, we see that RNAi knockdown of *sac1p* (human *SAC1P*) also significantly inhibited the MUV phenotype, however, to a lesser degree than knockdown of ORP5/8. Importantly, viability of organisms was not compromised upon *OSBPL5*, *OSBPL8*, or *SAC1P* gene silencing. Together, these results suggest that ER to PM PtdSer transport is required to support KRAS oncogenic signaling.

### Knockdown of ORP5 and ORP8 expression inhibits PtdSer transport to the PM and mislocalizes KRASG12V from the PM

To extend the *C. elegans* observations to mammalian cells, we used CRISPR/Cas9 to knock out (KO) ORP8 in CaCO-2 colorectal cancer cells (Fig 2A). ORP8 KO cells were then transfected with GFP-tagged oncogenic KRAS4B (GFP-KRASG12V) or a GFP-tagged PtdSer probe (GFP-LactC2). Intact basal PM sheets were prepared from these cells, labeled with GFP-antibodies coupled directly to 4.5-nm gold particles and visualized by EM (Hancock and Prior, 2005). We observed a significant decrease in anti-GFP immunogold labeling of both KRASG12V (Figs 2B and S1A) and LactC2 (Figs 2C and S1B) in ORP8 KO cells, indicating mislocalization from the inner PM. Spatial mapping analysis showed that the extent of clustering (*Lmax*) of KRASG12V remaining on the PM was also significantly decreased upon loss of ORP8 expression. Concordant with reduced KRASG12V PM binding and nanoclustering loss of ORP8 expression resulted in decreased MAPK signaling as measured by ppERK output (Fig 2A). To validate the mechanistic consequences of ORP8 KO, we measured the PM levels of PI4P, $PIP_2$, and $PIP_3$. To that end, we transfected CaCO-2 cells with GFP-tagged lipid probes and examined the extent of anti-GFP immunogold labeling by EM of intact PM sheets. The GFP-FAPP1-PH probe (FAPP1-PH) contains the pleckstrin homology (PH) domain of the FAPP1 protein, which selectively binds PI4P (Balla et al, 2005). The GFP-PLCδ-PH probe (PLCδ-PH) comprises the PH domain of PLCδ, which selectively binds $PIP_2$ (Hammond et al, 2012). The GFP-AKT-PH probe contains the PH domain of AKT which binds $PIP_3$ (Miao et al, 2010). In these EM experiments, we observed a significant increase in the amount of PI4P on the PM in ORP8 KO cells, which was further validated with GFP-P4M-SidM (Hammond et al, 2014), a second PI4P probe that exhibits a higher affinity for PI4P and has been shown to better detect P4P on the PM (Figs 2D and E

and S1C and D). Elevated PM levels of PI4P also correlated with significantly increased PM levels of both $PIP_2$ (Figs 2F and S1E) and $PIP_3$ (Figs 2G and S1F).

CaCO-2 cells do not form a well-organized confluent monolayer and are, thus, not well-suited for quantitative confocal microscopy analysis. We, therefore, used confluent MCF-7 breast cancer cells to visualize and quantify KRAS and PtdSer mislocalization. ORP5 and ORP8 were knocked down separately as well as simultaneously in MCF-7 cells (Fig 3A). Parental and knockdown cells were infected with bicistronic lentiviruses expressing either GFP-KRASG12V or GFP-LactC2 with mCherry-CAAX, a general endomembrane marker, and analyzed by confocal microscopy (Fig 3B). The extent of overlap between GFP and mCherry signals, indicative of colocalization, was quantified by Manders coefficients. The higher the Manders coefficient, the more extensive the colocalization of KRASG12V or LactC2 with endomembranes. These experiments showed that knockdown of ORP5 or ORP8 individually mislocalized KRASG12V and LactC2 from the PM to endomembranes to similar extents, with no discernible additive effect in double knockdown cells (Fig 3C).

### Depletion of PM PtdSer inhibits cell proliferation and anchorage-independent growth

We next evaluated the effects of ORP8 knockdown on KRAS and PtdSer localization in a panel of pancreatic cancer cell lines that are wild-type for KRAS, such as BxPC-3, or contain a KRAS mutation, such as PANC-1, MiaPaCa-2, and MOH. For each cell line, multiple stable monoclonal ORP8 knockdown (KD) cells were generated using shRNA lentiviral infection followed by puromycin selection. The use of multiple clones was designed to examine potential clonal variation after cell line selection. As in the model MCF-7 cell line, ORP8 knockdown caused significant mislocalization of both

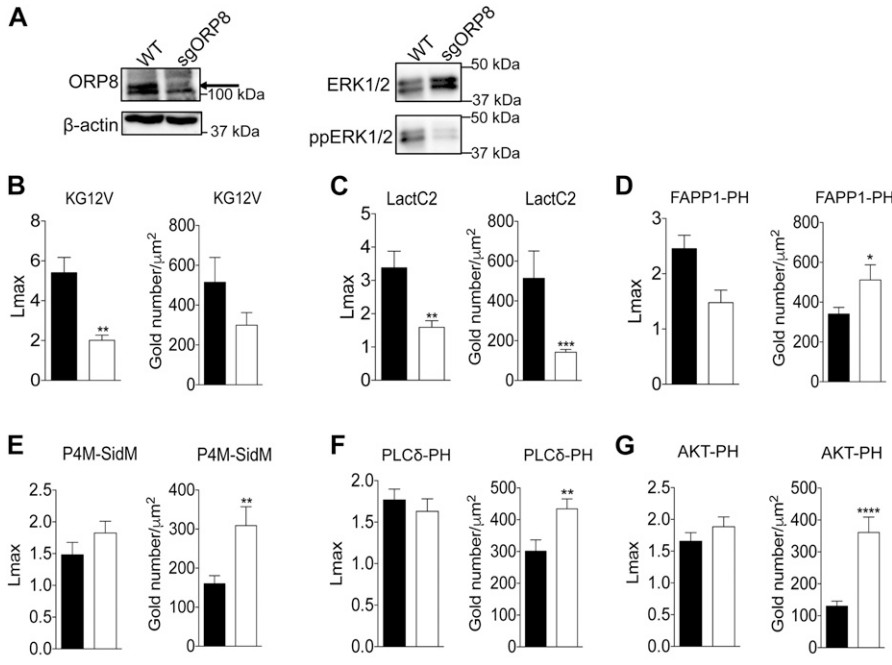

**Figure 2. KRAS and PtdSer, PI4P, $PIP_2$, and $PIP_3$ clustering and membrane localization changes in *OSBPL8* CRISPR KO CaCo-2 cells.**
**(A, B, C, D, E, F, G)** CRISPR/Cas9 KO of ORP8 in CaCo-2 cells was validated by Western blotting and MAPK signaling assayed as ppERK levels in Western blots. Total ERK and β-actin levels were used as loading controls. PM sheets prepared from CaCo-2 parental (WT) and sgORP8 cells transiently transfected with GFP-KRASG12V (B), GFP-LactC2 (C), GFP-FAPP1-PH (D), GFP-P4M-SidM (E), GFP-PLCδ-PH (F), or GFP-AKT-PH (G) were labeled with anti-GFP antibodies coupled directly to 4.5-nm gold particles and visualized by EM. The amount of KRASG12V, LactC2, FAPP1-PH, P4M-SidM, PLCδ-PH, and AKT-PH on the PM was measured as gold particle labeling per $\mu m^2$, and significant differences were quantified using *t* tests. Clustering of the GFP-tagged probes were quantified by univariate spatial analysis, summarized as *Lmax* values and significant differences were assessed using bootstrap tests (±SEM, n ≥ 12) (*$P < 0.05$, **$P < 0.01$, ***$P < 0.001$, ****$P < 0.00001$; KG12V: KRASG12V, LactC2: PtdSer probe, FAPP1-PH and P4M-SidM: PI4P probes, PLCδ-PH: $PIP_2$ probe, AKT-PH: PIP3 probe).

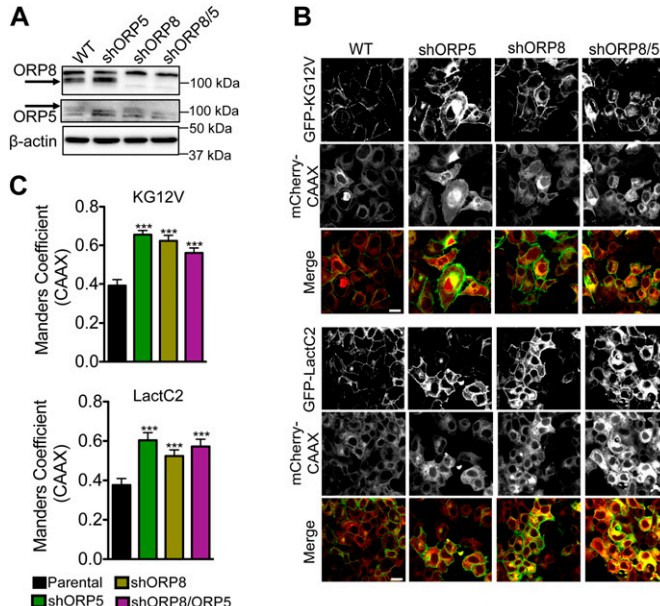

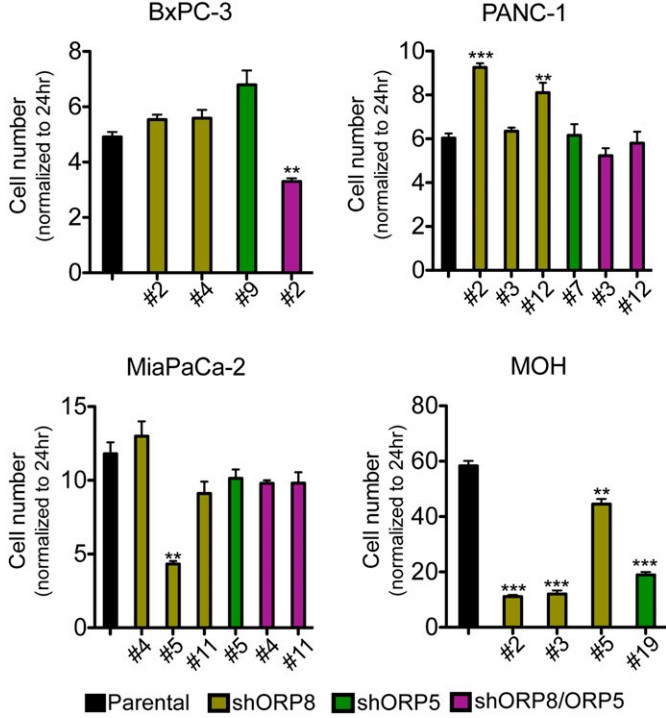

**Figure 3. Knockdown of ORP5 or ORP8 mislocalizes KRAS and PtdSer from the PM.**

**(A)** shRNA knockdown of ORP5 and ORP8 separately and simultaneously in MCF-7 breast cancer cells was validated by Western blotting, and β-actin levels were used as loading controls. **(B)** Parental (WT) and knockdown cells were transiently transfected with GFP-KRASG12V and mCherryCAAX (an endomembrane marker) or GFP-LactC2 and mCherryCAAX and imaged in a confocal microscope. Representative images are shown. **(C)** The extent of KRAS and LactC2 mislocalization was quantified using Manders coefficient, which measures the extent of colocalization/overlap of GFP and mCherry signals. Significant differences were evaluated using *t* tests (±SEM, n ≥ 6) (*$P < 0.05$, **$P < 0.01$, ***$P < 0.001$); scale bar 20 μm.

**Figure 4. ORP5 or ORP8 knockdown decreases the growth rate of KRAS-dependent pancreatic cancer cells**

ORP5 and ORP8 were knocked down separately and simultaneously by shRNA in BxPC-3, PANC-1, MiaPaCa-2, and MOH cells. Parental and knockdown cells were grown in six-well plates for 5 d and counted every day. Cell numbers of each cell line at day 5 were normalized to their cell number at 24 h and plotted. Significant differences were evaluated using *t* tests (±SEM, n = 3) (**$P < 0.01$, ***$P < 0.001$).

KRASG12V and LactC2 from the PM in each cell line tested, with the accumulation of both probes on endomembranes as visualized by confocal microscopy (Fig S2). We also generated monoclonal ORP5 knockdown cells using a similar protocol, as well as double knockdown of both ORP5 and ORP8 to assess possible compensation by one ORP homolog in the absence of the other (Fig S3).

First, we tested the effects on cell proliferation over the course of 5 d (Fig 4). In the case of the KRAS WT cell line BxPC-3, there was no discernible effect of knockdown of either ORP5 or ORP8 alone on proliferation rate. However, the simultaneous knockdown of both proteins modestly decreased the proliferation rate. In the case of PANC-1, which is a KRAS-independent cell line, that is, it is not addicted to oncogenic KRAS (Singh et al, 2009), ORP5 or ORP8 single knockdown resulted in cells growing significantly faster than parental cells; however, double knockdown clones grew more slowly. In contrast, single gene knockdown of either ORP homologue was sufficient to inhibit the proliferation of KRAS-dependent MiaPaCa-2 and MOH cells. In these proliferation assays, all MOH clones tested displayed a consistent response (Fig 4), whereas there was some variation in the behavior of different MiaPaCa2 ORP8 knockdown clones, including some that underwent senescence in culture and could not be analyzed further.

Anchorage-independent growth is a more stringent assessment of tumorigenic potential; therefore, we analyzed colony formation in soft agar (Fig 5). MiaPaCa-2 ORP5 and ORP8 KD clones showed

heterogeneous responses that correlated with their proliferation rate; MiaPaCa-2 KD clones that grew slower also showed reduced growth in soft agar. Double knockdown of both ORP proteins, however, had a more significant effect on anchorage-independent growth than on proliferation. Knockdown of either ORP protein completely abrogated colony formation in MOH cells. Interestingly, the clonal variation in MiaPaCa-2 ORP8 KD cells also correlated with the extent of mislocalization of LactC2 and KRASG12V as determined by confocal microscopy (Fig S4). Conversely, knockdown of ORP8 alone had no significant effect on the anchorage-independent growth of KRAS-independent PANC-1 cells. Of note, PANC-1 cells were more sensitive to knockdown of ORP5 than ORP8, whereas ORP8 knockdown more potently affected MiaPaCa-2 and MOH cells. As in the proliferation assays, double knockdown of both homologs had a stronger effect in all three KRAS-mutant transformed cell lines.

Analysis of signaling pathways revealed a paradoxical increase in MAPK signaling in nearly all KRAS-mutant ORP5 and ORP8 KD clones, evidenced as elevated ppERK1/2 levels. This likely indicates alleviation of the negative feedback on upstream components of the RAF-MEK-ERK pathway, in turn reflecting a reduction in the strength of KRAS signaling. One exception was PANC-1 KD cells which showed increased ppERK levels only when ORP5 was individually knocked down consistent with better response of these cells to ORP5 KD. Levels of pThr308AKT were increased in MiaPaCa-2 knockdown cells compared with parental and empty vector

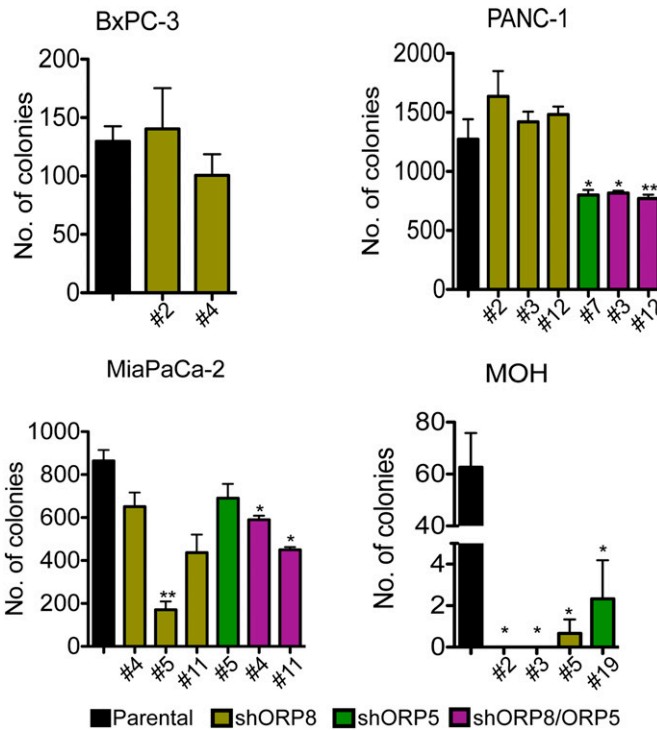

**Figure 5.  ORP5 or ORP8 knockdown decrease anchorage-independent growth capability of KRAS-dependent pancreatic cancer cells.**
ORP5 and ORP8 were knocked down separately and simultaneously by shRNA in BxPC-3, PANC-1, MiaPaCa-2, and MOH cells. Parental and knockdown cells were seeded in soft agar, with a base layer of 1% agar–media mixture and a top layer of a 0.6% agar–cell suspension mix in six-well plates. After 2–3 wk, the colonies were stained with 0.01% crystal violet and imaged. Colony numbers were quantified by ImageJ and significant differences were evaluated using $t$ tests (±SEM, n = 3) (*$P$ < 0.05, **$P$ < 0.01).

controls, presumably because of the accumulation of PI4P and, hence, PIP$_2$ on the PM after ORP knockdown (Fig 6), feeding into the PI3K/AKT pathway. In contrast, no pThr308AKT was detected in either parental or knockdown MOH cells. There was no significant difference in pThr308AKT levels between parental and knockdown cells in the PANC-1 and BxPC-3 lines. These results may possibly reflect different signaling outputs from the different KRAS point mutations in the cell lines (MiaPaCa-2: G12C, MOH: G12R, PANC-1: G12D) as well as KRAS dependency. pSer473AKT levels were elevated in KRAS-mutant ORP5 and ORP8 KD cells in all cell lines tested. Increased levels of YAP-1 and Epidermal Growth Factor Receptor (EGFR) activation have been previously reported to be compensatory mechanisms to KRAS inhibition in pancreatic cancer cells (Vartanian et al, 2013; Kapoor et al, 2014). Across the panel of KD cells, we saw considerable clonal variation with regard to YAP-1 and pEGFR levels (Fig 6) with no obvious correlation with KRAS mutational status. Finally, to evaluate whether KRAS mutational status affects ORP5 and ORP8 basal levels, we used the isogenic colorectal cancer cell line HCT116; the parental line harbors a heterozygous KRAS mutation, whereas its derivative line has a single WT KRAS allele after KO of the mutant allele via homologous recombination (Markowitz et al, 2009). ORP8 levels increased primarily in the derivative line upon ORP5 knockdown; however, the reverse was not observed (Fig S5).

### Inhibiting class III PI4Kα mislocalizes PtdSer and KRAS from the PM and selectively inhibits proliferation of KRAS-mutant pancreatic cancer cells

Currently, there are no available ORP5 or ORP8 inhibitors; therefore, we targeted the upstream component of PtdSer exchange: PI4KIIIα. Class III PI4Kα converts phosphatidylinositol (PI) to PI4P at the PM, which is then exchanged for PtdSer from the ER via ORP5 and ORP8 (Nakatsu et al, 2012; Clayton et al, 2013). The driving force of this process is a PI4P concentration gradient, which is kept high at the PM and low at the ER by Sac1 phosphatase which hydrolyzes PI4P back to PI (Fig 1A). The selective class III PI4Kα inhibitor compound 7 (C7) (Waring et al, 2014; Boura & Nencka, 2015; Raubo et al, 2015) should dissipate the PI4P concentration gradient between the PM and ER, functionally inhibiting ORP5 and ORP8. Because PI4KIIIα provides the driving force for both ORP5 and ORP8, inhibition should phenocopy a knockdown of both homologs.

MDCK cells stably expressing GFP-KRASG12V and mCherry-CAAX, or GFP-LactC2 and mCherry-CAAX were treated with C7 for 48 h and analyzed by confocal microscopy. Treatment with the inhibitor potently mislocalized both LactC2 and KRASG12V in a dose-dependent manner, with significant mislocalization seen at 30 µM (Fig 7A and B), consistent with the previously reported concentrations of C7 required to reduce cellular PIP$_2$ and PIP$_3$ levels (IC$_{50}$ = 30 µM) (Waring et al, 2014). To further quantify the amount of KRASG12V and LactC2 on the PM as well as the extent of nanoclustering after drug treatment, intact basal PM sheets of MDCK cells were labeled with gold-conjugated anti-GFP antibodies 48 h after treatment with 30 µM of C7 and analyzed by EM. C7 treatment caused significant mislocalization of both KRASG12V and LactC2 from the PM and decreased their nanoclustering (Fig 7C).

Finally, we tested the effects of C7 on cell proliferation in a panel of pancreatic cancer cell lines as well as the immortalized pancreatic cell line HPNE. Ten different concentrations of C7 were tested ranging from 1 nM to 30 µM. The results show that C7 had no effect on the non-transformed cell line HPNE. All transformed cells were sensitive to C7, but the calculated IC$_{50}$ values for growth inhibition were much lower for KRAS mutant than KRAS WT cells, with the most sensitive lines being the KRAS-dependent MOH and MiaPaCa-2 lines (Fig 7D).

### Expression of ORP5 and ORP8 in KRAS-mutant cancers

High ORP5 expression has been previously shown to be associated with poorer survival rates in patients with pancreatic adenocarcinoma (PDAC) (Koga et al, 2008). To further investigate whether this is linked to oncogenic KRAS signaling, we analyzed *OSBPL5* mRNA expression and *KRAS* mutation data in GDC (Genomic Data Commons) TCGA-PAAD, TCGA-LUNG, and TCGA-PANCAN, cohorts of pancreatic adenocarcinoma, non–small cell lung cancer, and a pan-cancer cohort comprising 33 different cancer types, respectively (Goldman et al, 2019 Preprint). In all three cohorts, ORP5 expression was significantly up-regulated in *KRAS*-mutant subgroups compared with *KRAS* wild-type subgroups (Fig 8A). Additional analyses showed that increased expression of ORP5 or ORP8 correlate with shorter overall survival times for patients in all three cohorts (Fig 8B and C).

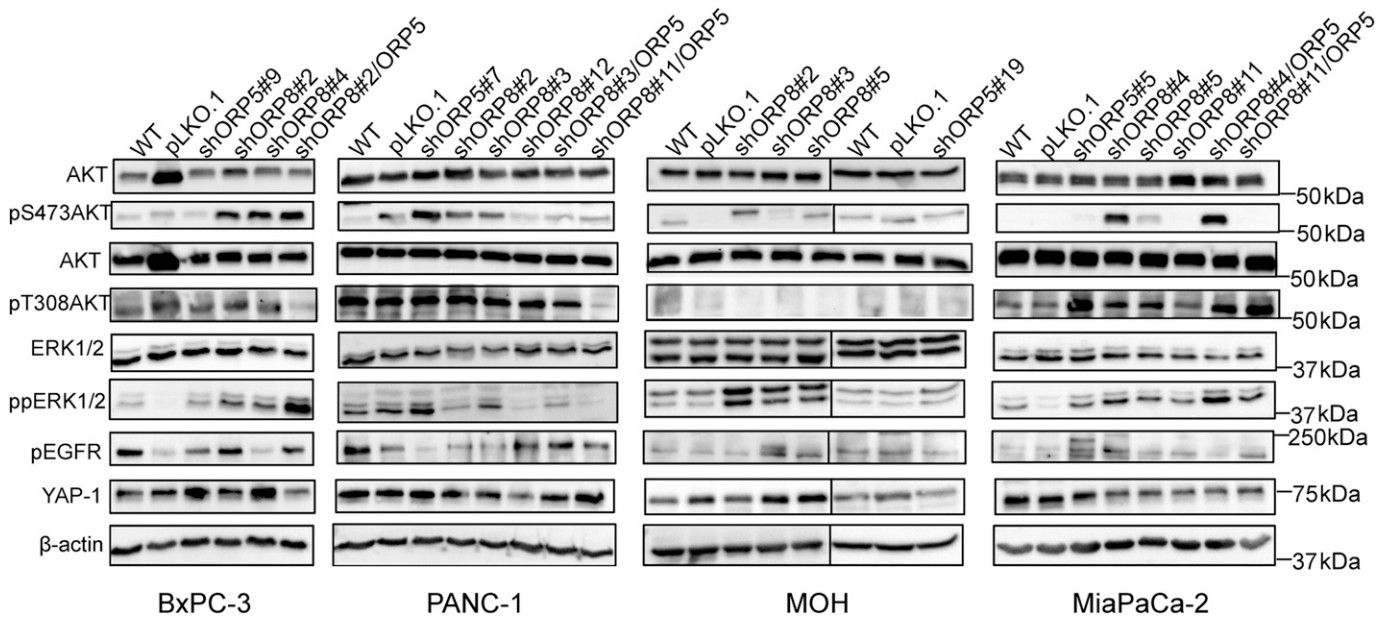

**Figure 6. Consequences of ORP5 and ORP8 knockdown on downstream MAPK and PI3K/AKT signaling.**
Protein from BxPC-3, PANC-1, MiaPaCa-2, and MOH parental, single and double ORP knockdowns as well as cells transfected with empty vector control (pLKO.1) were harvested, and 20 μg was subjected to SDS–PAGE and used for Western blotting. EGFR, MAPK, and PI3K signaling were assayed as pEGFR, ppERK, and pAKT levels, respectively. Amplification of YAP-1 was also evaluated. Total ERK, total AKT, and β-actin levels were used as loading controls.

## Discussion

We show here that maintenance of PM PtdSer levels is absolutely required to maintain KRAS PM localization and hence oncogenic function. Thus, knockdown or inhibition of any component of the ORP5/8 ER to PM PtdSer transport process abrogates KRAS function in multiple cells and organisms. First, RNAi silencing of orthologs of

ORP5, ORP8, or the ER PI4P phosphatase, Sac1, inhibited oncogenic let-60 (KRAS) signaling in *C. elegans*. Second, shRNA knockdown or CRISPR/Cas9 KO of ORP5 or ORP8 mislocalized PtdSer and KRAS from the PM and decreased the extent of KRAS PM clustering in human pancreatic, breast and colorectal cancer cells. ORP5 or ORP8 knockdown concordantly inhibited the proliferation and anchorage-independent growth of KRAS-dependent pancreatic

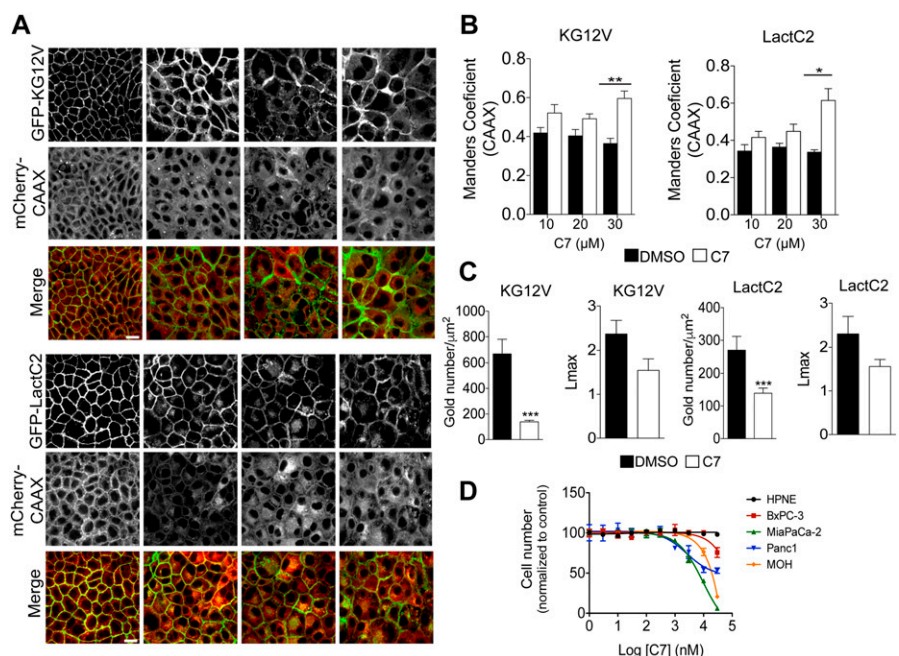

**Figure 7. PI4KIIIα inhibition mislocalizes KRAS and PtdSer from PM and inhibits growth of KRAS-dependent pancreatic cancer cells.**
**(A)** MDCK cells stability expressing GFP-KG12V and mCherryCAAX or GFP-LactC2 and mCherryCAAX were treated with DMSO or the PI4KIIIα inhibitor C7 for 72 h at varying concentrations, and then imaged with confocal microscopy. Representative images are shown. **(B, C)** The extent of KRAS and LactC2 mislocalization was quantified using Manders coefficient, which measures the extent of colocalization/overlap of GFP and mCherry signals. Significant differences were quantified using *t* tests (±SEM, n ≥ 5) (C) Basal PM sheets from MDCK cells in (A) treated with 30 μM of C7 for 48 h were prepared and labeled with anti-GFP antibodies coupled directly to 4.5-nm gold particles and visualized by EM. The amount of KRASG12V and LactC2 on the PM was measured as gold particle labeling per μm², and significant differences were quantified using *t* tests. KRAS and LactC2 clustering were quantified by univariate spatial analysis, summarized as *Lmax* values and significant differences were assessed using bootstrap tests (±SEM, n ≥ 12).
**(D)** HPNE, BxPC-3, MiaPaCa-2, PANC-1 and MOH cells were seeded in 96-well plates. After 24 h, fresh growth medium supplemented with 1% DMSO or increasing C7 concentrations were added and the cells were allowed to grow for another 72 h and then counted (±SEM, n = 3) (*P < 0.05, **P < 0.01, ***P < 0.001; KG12V: KRASG12V, LactC2: PtdSer probe), scale bar 20 μm.

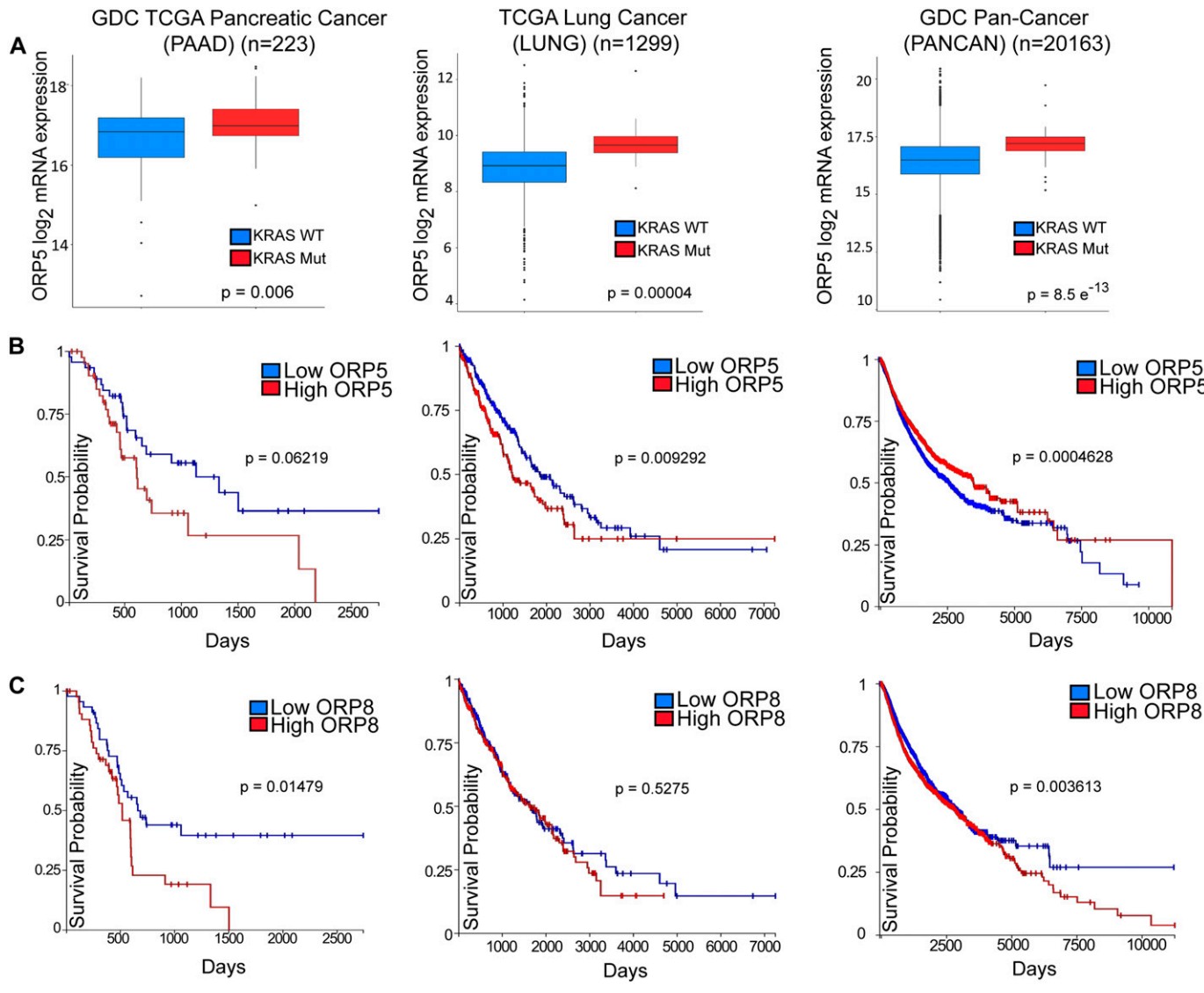

**Figure 8. High expression levels of *OSBPL5* and *OSBPL8* correlate with poorer prognosis in cancer patients.**
**(A, B, C)** Box plots indicating quartiles of ORP5 mRNA expression level in patient samples in cohorts of pancreatic cancer (GDC TCGA PAAD, n = 223), lung cancer (TCGA LUNG, n = 1,299), and of 33 types of cancer (GDC Pan-Cancer [PANCAN], n = 20,163) with or without *KRAS* mutations. Statistical significance was analyzed with Welch's *t* test. Kaplan–Meier survival plots based on expression levels of ORP5 (B) and ORP8 (C) in cohorts listed in (A). Plots were generated using the University of California, Santa Cruz (UCSC) Xena Browser.

cells. Third, inhibiting PI4KIIIα also reduced PtdSer and KRAS PM levels sufficiently to selectively abrogate the growth of KRAS-dependent pancreatic cancer cells. Finally, these observations have some clinical correlates in that PDAC patients with higher expression levels of ORP5 or ORP8 have poorer clinical outcomes, and more generally, *KRAS* mutational status is associated with higher ORP5 expression across multiple cancer types. These clinical correlates are consistent with selection for more robust maintenance of PM PtdSer levels to support KRAS oncogenesis.

Recent work suggests that ORP5 is primarily responsible for PtdSer and PI4P trafficking at ER-PM MCSs and that ORP8 may only be recruited to these sites upon PIP$_2$ accumulation. Under conditions of PIP$_2$ accumulation, ORP8 can also exchange PIP$_2$ for PtdSer (Sohn et al, 2018). Therefore, knocking down ORP5 would lead to an accumulation of PI4P at the PM and, hence, an increase in

PIP$_2$ levels, which would lead to ORP8 recruitment to the PM and at least partial restoration of PtdSer PM levels by the dual action of ORP8 as a PtdSer/PI4P and PtdSer/PIP$_2$ exchanger (Ghai et al, 2017). This would explain why we did not observe an increase of ORP5 in ORP8 HCT116 knockdown cells but observed an increase of ORP8 when ORP5 was knocked down (Sohn et al, 2018). Whether PIP$_2$ is more important than PI4P for PtdSer transport to the PM and cannot be compensated for remains to be determined. However, the importance of each homolog seems to be cell line specific, as we noticed that PANC-1 cells were more sensitive to ORP5 knockdown, whereas MiaPaCa-2 cells were more sensitive to ORP8 knockdown. ORP5/8 has been reported to also function at ER–mitochondria MCSs, and knockdown of either ORP leads to altered mitochondrial morphology and reduced oxygen consumption (Galmes et al, 2016). However, it appears that ORP5 interacts more

extensively with mitochondrial outer membrane proteins, so it might play a more important role than ORP8 here. It has also been reported that KRAS-independent lines (Ribosomal S6 Kinase [RSK]-dependent) depend on oxidative phosphorylation and have increased reactive oxygen species, whereas KRAS-dependent lines rely on glycolysis (Galmes et al, 2016; Yuan et al, 2018). Together, this may be a possible explanation for the increased sensitivity of PANC-1 cells to ORP5 knockdown. Nonetheless, all cell lines were more sensitive to the simultaneous knockdown of both ORP5 and ORP8 compared with either one alone. One potential method to partially sustain PtdSer levels on the PM in double knockdown cells may be through increased vesicular transport of PtdSer through recycling endosomes, which are enriched with this phospholipid (Matsudaira et al, 2017). In a recent article, Venditti et al (2019) showed that ORP10 transfers PtdSer from its site of synthesis at the ER to the trans-Golgi (TGN) and that TGN PtdSer levels correspond to the activity levels of phosphatidylserine synthase I in the ER (Venditti et al, 2019). They also showed that ORP10 knockdown resulted in significant reduction of Golgi PtdSer levels, but no change in PM PtdSer levels. To this end, we would speculate that the Golgi pool of PtdSer does not contribute, at least in a noticeable way, to the PM levels of PtdSer and, hence, of KRAS. However, we still do see PtdSer and KRAS located on the PM of ORP5/ORP8 double knockdown MCF-7 cells as well as in C7-treated MDCK cells. Thus, PtdSer that accumulates in the ER because of ORP5/8 inactivation may be shuttled to the TGN via ORP10 and transported to the PM via vesicular transport. However, because the PM levels of KRAS and PtdSer after ORP5/8 depletion are still significantly lower than in parental cells, we would again conclude that the Golgi pool of PtdSer is not a major contributor to KRAS PM localization. Further investigation into how inhibition of ORP5 or ORP8 affects vesicular transport is warranted.

Sohn et al (2018) also reported that PI4KIIIα inhibition decreased the amount of ORP5 and ORP8 localized to the PM (Sohn et al, 2018). In addition, they showed that prolonged overexpression of PIP5K1b, which converts PI4P to PIP$_2$, caused a redistribution of PI4P to endosomal and Golgi compartments, effectively reducing PI4P PM levels and, therefore, PIP$_2$ PM levels by limiting substrate availability. This in turn led to a reduction of PtdSer levels on the PM, further validating the rationale of targeting PI4KIIIα to reduce PI4P, and hence PtdSer PM levels to inhibit KRAS signaling. Concordantly, we showed that treating MDCK cells with the selective class III PI4Kα inhibitor, C7, resulted in redistribution of PtdSer, and hence KRAS, from the PM to endomembranes in a dose-dependent manner. Our results are also in accordance with others who found a 50% reduction in overall PtdSer levels and a depletion of PtdSer levels at the PM in response to PI4KIIIα inhibition or PI4KIIIα KO (Chung et al, 2015; Sohn et al, 2016). The decrease of PtdSer synthesis was due to the accumulation of PtdSer at the ER, and its consequent negative feedback inhibition on PtdSer synthase I and II.

Importantly, we also found that C7 selectively inhibited the proliferation of pancreatic cancer cell lines, with KRAS-dependent lines displaying increased sensitivity compared with KRAS-independent lines. KRAS wild-type cells were only affected at high concentrations and immortalized normal pancreatic cells were unaffected even at the highest concentrations tested. Previously, PI4KIIIα inhibition was found to increase radiosensitivity in diverse cancer cell lines in vitro as well as in immune-competent and nude

mouse models of breast and brain cancer (Park et al, 2017). Thus, targeting PI4KIIIα is tolerated in these animals at concentrations that result in antitumor effects. Furthermore, there are no reports of activating mutations or deletions of PI4Ks in cancer, allowing us to be cautiously optimistic of a low mutability rate of this gene (Clayton et al, 2013). Finally, PI4KIIIα inhibition decreased pAKT levels by decreasing PIP$_3$ PM levels in breast cancer cells; hence, it can act in KRAS-mutant cells as a target to simultaneously inhibit both MAPK and PI3K pathways, which is an attractive notion given that 93% of *PIK3CA* mutations in PDAC co-occur with *KRAS* mutations (Park et al, 2017; Waters & Der, 2018). Therefore, PI4KIIIα has merit as a novel treatment target for KRAS-dependent tumors that warrants further research.

Koga et al (2008) showed that PDAC patients with high ORP5 expression had a 36.4% 1-yr survival rate, whereas those with low ORP5 expression had a one-year survival rate of 73.9%. In addition, they showed that siRNA knockdown of ORP5 decreased the invasion potential of pancreatic cancer cells in matrigel, whereas overexpression of ORP5 increased invasion. This correlated in patients, whereby high ORP5 expression corresponded with invasion of cancer cells to the main pancreatic duct, leading to early relapse (Koga et al, 2008; Ishikawa et al, 2010). Through analyses of other patient cohorts in the TCGA (The Cancer Genome Atlas) database, we found that this negative correlation between ORP5 expression level and overall survival held true not only in PDAC but also in multiple other cancer types. We also observed significantly increased ORP5 expression in KRAS-mutant tumors compared with KRAS wild-type tumors. We also found that high expression of ORP8 also correlated with decreased patient survival in multiple cancers including PDAC.

In conclusion, we have shown that targeting ORP5, ORP8, or PI4KIIIα depletes the PM of PtdSer resulting in mislocalization of KRAS and reduced nanoclustering of KRAS that remains PM bound in all cell lines tested. In turn, these perturbations of KRAS PM interactions lead to reduced proliferative and tumorigenic capacity of KRAS-mutant cancer cells. To the best of our knowledge, this is the first study to establish the mechanistic connection between ORP5 and ORP8 with KRAS and their important role in KRAS-driven cancers. Our results with ORP5 and ORP8 knockdown as well as with PI4KIIIα inhibition have possible implications for cancer therapy in KRAS-mutant tumors. In sum, we have demonstrated that reducing PM PtdSer levels is selectively toxic to KRAS-transformed cells and that there was no organismal toxicity associated with blocking PtdSer ER to PM transport in *C. elegans*, whereas activated let-60 signaling was suppressed. The exquisite binding specificity of the KRAS membrane anchor for PtdSer, which is essential for PM targeting, is thus a vulnerability in KRAS-mutant tumors that may be amenable to therapeutic exploitation.

# Materials and Methods

## Materials

Class III PI4K alpha inhibitor Small Molecule (Tool Compound), C7, was purchased from Cancer Research UK (ximbio.com, cat. no. 153579, distributed by Ximbio) and dissolved in DMSO. Cell culture

media were purchased from HyClone and GIBCO. FBS was purchased from GIBCO. Puromycin was purchased from Thermo Fisher Scientific (BP2956-100). Anti-phospho-p44/42 MAPK (ERK1/2) Thr202/Tyr204 (43702), total ERK1/2 (4659S), p-c-Raf (9427S), phospho-Ser473 AKT (4060L), phospho-Thr308 AKT (9257S) pan-AKT (2920S), phospho-EGFR (4407L), GFP (2956S), and $\beta$-Actin (A1978) antibodies were purchased from Cell Signaling Technology. Anti-osbpl5 (NB100-57071) and YAP-1 (NB110-58358) antibodies were purchased from Novus. Anti-osbpl8 (ab99069) antibody was purchased from Abcam. Rabbit anti-mGFP antibodies for immunogold labeling were generated in house. Agarose-low melting point (CAS 39346-81-1) was purchased from Sigma-Aldrich.

## Cell lines

MDCK, HPNE, and PANC-1 cells were purchased from American Type Culture Collection. TLA293T cells were a generous gift from Dr Guangwei Du, McGovern Medical School, Houston, TX. BxPC3, MOH, and MiaPaCa-2 were kindly provided by Dr Craig Logsdon at MD Anderson Cancer, Houston, TX. *KRAS* (+/−) HCT116 isogenic cell line pair was kindly provided by Dr Scott Kopetz at MD Anderson Cancer Center, Houston, TX. MDCK, PANC-1, and TLA293T cells were grown in DMEM supplemented with 2 mM L-glutamine and 10% FBS. HPNE cells were cultured in 75% DMEM and 25% Medium M3 Base supplemented with 5% FBS, 10 ng/ml human recombinant EGF, 5.5 mM D-glucose, and 750 ng/ml puromycin. MiaPaCa-2 cells were cultured in DMEM supplemented with 2 mM L-glutamine and 10% FBS and 2.5% horse serum. BxPC3 and MOH cells were cultured in RPMI-1640 medium supplemented with 2 mM L-glutamine and 10% FBS. HCT116 cells were cultured in RPMI 1640 including 2 mM L-glutamine and 25 mM sodium bicarbonate, supplemented with 10% FBS. All cell lines were grown at 37°C in 5% $CO_2$.

## Western blotting

Cells were washed in cold PBS and lysed in buffer containing 50 mM Tris-Cl (pH 7.5), 75 mM NaCl, 25 mM NaF, 5 mM MgCl$_2$, 5 mM EGTA, 1 mM dithiothreitol, 100 $\mu$M NaVO4, and 1% NP40, in addition to protease inhibitors. Whole cell lysates (20 $\mu$g) were immunoblotted and signals were detected with enhanced chemilumisescence (Thermo Fisher Scientific) and quantified in a LumiImager (Roche Molecular Biochemicals).

## Identification of OSBPL5 and OSBPL8 homologs/orthologs in *C. Elegans*

FASTA sequences for human OSBPL5 and OSBPL8 were obtained from the NCBI protein database. Subsequently, using the blast tool in WormBase (https://wormbase.org/tools/blast_blat), homologs/orthologs of OSBPL5 and OSBPL8 were identified. Hits with a percentage identity of 30 and above were considered as candidate genes.

## *C. elegans* vulva quantification assay

RNAi-mediated knockdown of osbpl5 and osbpl8 was induced by feeding *let-60*(n1046) worms with *Escherichia coli* HT115 generating dsRNA to target genes from their L1 stage to adult stage. A DIC (Differential Interface Contrast)/Nomarski microscope was used to score the MUV phenotype.

## Generation of CRISPR/Cas9 cell line

CaCO-2 cells were transduced with *OSBPL8* sgRNA (3′-TGCAAA-TCTTTGGTTGGCGT-5′) plus Cas9 expression followed by puromycin selection (4 $\mu$g/ml) after 24 h. Single colonies were generated from the pool of polyclonal KO cells.

## Lentiviral transduction

For lentivirus production, TLA293T cells were transfected with ViraPower lentiviral packaging mix (K4975-00) using Lipofectamine (18324-012) and PLUS Reagent (10964-021). All reagents were purchased from Invitrogen. Lentiviral particles were collected 48 and 72 h after transfection, and then concentrated with Lenti-X concentrator (931232; Clontech). Titers were estimated with Lenti-X Go-Stix (#631244; Clontech).

## shRNA knockdown and bicistronic transient infection

OSBPL5 shRNA was purchased from Sigma-Aldrich (cat. no. SHCLNG-NM_020896, shRNAa: NM_020896.2-1316s1c1: 3′-CCGGGAA-CAAGCTCTCCGACTACTACTCGAGTAGTAGTCGGAGAGCTTGTTCTTTTTTG-5′, shRNAb: NM_020896.2-2980s1c1: 3′-CCGGGCCTTAATGCTAAAGC-CAAATCTCGAGATTTGGCTTTAGCATTAAGGCTTTTTTG-5′, and shRNAc: NM_020896.2-2732s1c1: 3′- CCGGGTTCATTAACCACATCCTCAACTCGA-GTTGAGGATGTGGTTAATGAACTTTTTTG-5′). shRNA constructs for OSBPL5 were cloned into pLenti6.3-V5-TOPO vector (K5315-20; Invitrogen). OSBPL8 shRNA pre-packaged into transduction particles was purchased from Dharmacon (cat. no. V3SH7602-226843976: 3′-TGACAAGCCTATAAACACC-5′). The empty pLKO.1-TRC cloning vector was a gift from David Root (plasmid #10878; Addgene). GFP-KG12V/mCherry-CAAX and GFP-LactC2/mCherry-CAAX bicistronic plasmids were generated in-house and packaged into lentiviral particles. Pancreatic cancer cells were seeded at 4 × 10$^5$ cells per well in six-well plates, infected with lentiviral particles 24 h later at 70% confluency, and osbpl5 and osbpl8 knockdown stable cell lines of were selected for with puromycin (4 $\mu$g/ml).

## Confocal microscopy

Cells were seeded onto coverslips and allowed to grow for 48 h before fixation with 4% PFA and quenching with 50 mM NH$_4$Cl. Coverslips were then mounted in Mowiol and visualized by confocal microscopy (Nikon A1R) using a 60X objective.

## EM and spatial mapping

Basal PM sheets of CaCO-2 and MDCK cells were prepared, fixed with 4% PFA and 0.1% glutaraldehyde, and labeled with affinity-purified anti-GFP antisera conjugated to 4.5-nm gold as described previously (Prior et al, 2003). Digital images of immunolabeled membrane sheets were taken with a transition electron microscope at 100,000× magnification and intact 1-$\mu m^2$ areas were identified with ImageJ. (*x, y*) coordinates of the gold particles were determined

as described in Prior et al (2003). Univariate K function (Ripley, 1977) was calculated and standardized on a 99% confidence interval (Diggle et al, 2000; Hancock and Prior, 2005; Prior et al, 2003), whereby an *L(r)-r* value greater than the confidence interval is indicative of significant clustering. The extent of clustering is represented by the (*Lmax*) value, the maximum value of the K function. Bootstrap tests were used to analyze differences between replicated point patterns as described previously (Diggle et al, 2000), and statistical significance was determined by evaluation against 1,000 bootstrap samples.

### Proliferation assay

For shRNA knockdown studies, BxPC-3, PANC-1, MiaPaCa-2, and MOH parental and knockdown cells were seeded at a density of $2 \times 10^5$ cells/well in six-well plates and counted every day for 5 d using the countess automated cell counter (Invitrogen). For drug treatment studies as validated by Raubo et al (2015), HPNE ($5 \times 10^3$), BxPC-3 ($4 \times 10^3$), PANC-1 ($4 \times 10^3$), MiaPaCa-2 ($2 \times 10^3$), and MOH ($1.5 \times 10^3$) cells were seeded in 96-well plates. After 24 h, fresh growth medium supplemented with 1% DMSO or differing drug concentrations were added, and the cells were allowed to grow for another 72 h. Cell numbers were determined by CyQuant Proliferation Assay (Thermo Fisher Scientific) according to the manufacturer's protocol.

### Anchorage-independent growth assay

BxPC-3 ($10 \times 10^3$), PANC-1 ($5 \times 10^3$), MiaPaCa-2 ($5 \times 10^3$), and MOH ($5 \times 10^3$) parental and knockdown cells were seeded in soft agar in six-well plates, with a base layer of 1% agar–media mixture, and a top layer of 0.6% agar–cell suspension mix as performed in (Borowicz et al, 2014). After 2–3 wk, colonies were stained with 0.01% crystal violet and imaged. Colony numbers were quantified by ImageJ.

### Bioinformatic analysis using UCSC Xena browser

ORP5 and ORP8 mRNA expression and *KRAS* mutational status in patients and their overall survival were analyzed and visualized using data in GDC TCGA-PAAD, TCGA-LUNG, and GDC-PANCAN, by Xena browser (https://xenabrowser.net/) (Goldman et al, 2019 *Preprint*).

### Statistical analysis

Results are presented as the mean ± SEM. Prism version 5.0 (GraphPad Software) was used for two-tailed *t* test. Levels of significance are labeled as: *$P < 0.05$; **$P < 0.01$; ***$P < 0.001$; ****$P < 0.0001$.

## Supplementary Information

## Acknowledgements

This Research was supported by Cancer Prevention and Research Institute of Texas (CPRIT) grant RP170233 to JF Hancock.

### Author Contributions

WE Kattan: conceptualization, formal analysis, validation, investigation, visualization, methodology, and writing—original draft, review, and editing.
W Chen: validation and investigation.
X Ma: investigation.
TH Lan: investigation.
D van der Hoeven: investigation.
R van der Hoeven: investigation.
JF Hancock: conceptualization, supervision, funding acquisition, project administration, and writing—review and editing.

### Conflict of Interest Statement

The authors declare that they have no conflict of interest.

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
