## [Reviewer comments · Life Science Alliance]

Life Science Alliance

Targeting Plasma Membrane Phosphatidylserine Content to Inhibit Oncogenic KRAS Function

Walaa Kattan, Wei Chen, Xiaoping Ma, Tien Hung Lan, Dharini van der Hoeven, Randsome van der Hoeven, and John Hancock

DOI: <https://doi.org/10.26508/lsa.201900431>

Corresponding author(s): John Hancock, University of Texas Health Science Center at Houston

Review Timeline:

Submission Date:	2019-05-17
Editorial Decision:	2019-06-04
Revision Received:	2019-08-10
Editorial Decision:	2019-08-16
Revision Received:	2019-08-16
Accepted:	2019-08-19

Scientific Editor: Andrea Leibfried

Transaction Report:

June 4, 2019

Re: Life Science Alliance manuscript #LSA-2019-00431-T

Prof. John F. Hancock
University of Texas Health Science Center at Houston
Integrative Biology and Pharmacology
6431 Fannin Street, MSB 4.098
Houston, Texas 77030

Dear Dr. Hancock,

Thank you for submitting your manuscript entitled "Targeting Plasma Membrane Phosphatidylserine Content to Inhibit Oncogenic K-Ras Function" to Life Science Alliance. The manuscript was assessed by expert reviewers, whose comments are appended to this letter.

As you will see, the reviewers appreciate your analyses and provide constructive input on how to further strengthen your work. I would thus like to invite you to submit a revised version of your manuscript to us, addressing the individual points raised by the reviewers. Such a revision seems straightforward, but please get in touch with me in case you would like to discuss individual revision points further.

I hope that the comments below will prove constructive as your work progresses.

Thank you for this interesting contribution to Life Science Alliance. We are looking forward to receiving your revised manuscript.

Sincerely,

Andrea Leibfried, PhD
Executive Editor

Life Science Alliance
Meyerhofstr. 1
69117 Heidelberg, Germany
t +49 6221 8891 502
e a.leibfried@life-science-alliance.org
www.life-science-alliance.org

B. MANUSCRIPT ORGANIZATION AND FORMATTING:

Reviewer #1 (Comments to the Authors (Required)):

In their manuscript entitled 'Targeting Plasma Membrane Phosphatidylserine Content to inhibit oncogenic K-RAS function' Hancock and co-workers manipulate the PtdSer/ PtdIns(4)P cycle at ER-PM membrane contact sites to reduce the PtdSer dependent association of KRAS with the plasma membrane, and thus reduce its oncogenic potential. By combining experiments in nematodes and human cells, Hancock and coworkers show that silencing the expression of PtdSer/ PtdIns(4)P transfer proteins ORP5 and ORP8 or by reducing the activity of PI4KIIIa they can reduce

KRAS recruitment at the PM its local clustering and, as a consequence the activation of downstream signaling pathways. The central hypothesis of this manuscript (i.e., that PtdSer on PM depend on the activities of ORP5, ORP8 and PI4KIIIa, and that manipulating PtdSer levels at the PM one can inhibit KRAS) is supported by solid literature produced by the authors and by others. The effects of the manipulations used in this study on PtdSer levels at the PM and on KRAS localization are convincing and the methodologies used to measure them are appropriate. Nonetheless the experiments aimed at testing the effects of ORP5 and ORP8 loss of function on PtdIns(4)P and of PtdIns(4,5)P₂ localization and levels require some substantiation. Also, in general it would be good to have some representative images of the localization experiments to accompany the quantitation expressed in graphs. The manuscript is generally well written though it is sometimes cryptic and in some specific parts not accurate enough (see the specific comments below). In my opinion the authors need to address these issues in order for their manuscript to attain a sufficient quality for publication in LSA.

Specific comments

1. In Figure 2D the authors use a PtdIns(4)P reporter FAPPH (most likely FAPP1-PH-GFP) to quantitate the PM levels of PtdIns(4)P in ORP8-KO cells. FAPP1-PH in mammalian cells specifically recognizes the Golgi complex pool of PtdIns(4)P showing little if any localization to the PM (Godi et al. NCB 2004). More appropriate tools to evaluate the PM PtdIns(4)P levels are the P4M probe (Hammond et al JCB 2014) or the anti- PtdIns(4)P antibody used as described in Hammond Biochem. J. 2009.
2. Again on Figure 2D, the methodology described by Hancock and Prior (2005) is very powerful in assessing the nanoclustering at the PM, nonetheless this represents a second level approach and it has to be coupled to more standard imaging techniques (i.e., confocal microscopy) in order for the authors to conclude that their manipulations induce significant changes in the localization of their probes. The authors follow this rule when addressing the effects of their manipulations on KRAS (KRASG12V-GFP) and PtdSer (LactC2-GFP) localization, they should do so also when it comes to assess the localization of FAPP1-PH-GFP or PLCd (again here most likely PLCd-PH-GFP).
3. In a recent publication Venditti et al. showed that PtdSer is transported by the lipid transfer protein ORP10 to the trans Golgi complex from where most probably it gets transported to the PM (Venditti et. Al JCB 2019). What is the contribution of this additional PtdSer pool to the recruitment of KRAS to the PM (or to endo-membranes)?
4. The authors hypothesize that the effects they observe on AKT phosphorylation (pSer-473) might depend on the increased metabolic flux on PM from PtdIns(4)P to PtdIns(4,5)P₂ to PtdIns(3,4,5)P₃. This prediction needs to be corroborated by data aimed at assessing the actual levels of PtdIns(3,4,5)P₃ at the PM (i.e., by the use of AKT-PH-GFP as a reporter). Also the use of the anti- AKT pSer-473 antibody here is a bit confusing as phosphorylation in this specific serine can occur as a result of a feedback loop mechanism. I would suggest to use the anti AKT pThr-308 as a more direct measure of AKT activation downstream of PtdIns(3,4,5)P₃ production.
5. Some of the tools used in this study are not described in a sufficient detail. The FAPPH and PLCd probes, for instance are not described and there is no clear reference in the paper to the literature where one can find a more systematic description.

Reviewer #2 (Comments to the Authors (Required)):

ORP5 and ORP8 transfer the anionic lipid phosphatidylserine (PS) from the ER to the plasma membrane, owing to the ability of their ORD domain to extract PS and counter-exchange it for PI4P.

The small GTPase KRAS interacts with negatively charged membranes, owing to complex electrostatic interactions between its C-terminal polybasic and prenylated tail. These two elementary properties suggest that the plasma membrane localization of KRAS might depend on the activity of ORP5/8. This paper makes and addresses this prediction. Manipulating ORP5 or ORP8 or interfering with the enzymes that feed their cycle, including a PI4 kinase and a PI4P phosphatase, impedes the recruitment of KRAS at the plasma membrane. Overall, this manuscript is solid and well executed. The points below might be addressed by text clarification and/or additional experiments.

#1. Abstract: "Inhibiting KRAS function through regulating PM lipid PtdSer content may represent a viable strategy for KRAS-driven cancers". Isn't this final statement too optimistic, given the pleiotropic effects of plasma membrane electrostatics on various events, including cytoskeleton attachment, clathrin mediated endocytosis, numerous signaling cascades? It is unlikely that KRAS is the only protein sensitive to PS content, notably in non-cancerous cells.

#2. C elegans experiments (Figure 1B). The results are clear cut: knocking down obr2 and to a lesser extent obr4 strongly diminishes a phenotype (MUV) that is caused by the hyperactivity of KRAS. However, the sentences accompanying these results are difficult to follow. As stated by the authors, there is no clear homolog or orthologs of ORP5 and ORP8 in C elegans. However, blast analysis suggest that three genes encode for proteins related to ORP5/8: obr2, obr3 and obr4. The authors say that inhibiting the expression of any of these genes affect the MUV phenotype; yet in Figure 1B they show only the results for obr2 and obr4. In addition, in a following sentence, they refer to ORP5 and ORP8, not to obr: "Knockdown of either ORP5 or ORP8 expression potently suppressed the MUV phenotype, with around 90% of the population displaying a single vulva, similar to treatment with the MEK inhibitor U0126 as well as to the positive controls riok-1 and hoe-1, previously described as potent suppressors of the MUV phenotype". Please clarify the corresponding paragraph.

#3. The Western Blot shown in Fig 2A is not convincing.

#4. Please better justify the transition between the underlying strategies used in Figure 2 and Figure 3. In Figure 2, the authors used CRISPR/Cas9 to knock out ORP8 in CaCO-2 colorectal cancer cells. In Figure 3, the authors use interference RNA to knock down ORP5, ORP8 or both in MCF-7. The sentence accompanying this transition is "To further visualize KRAS and PtdSer mislocalization, ORP5 and ORP8 were knocked down separately as well as simultaneously in MCF-7 breast cancer cells (Figure 3A)." Why changing the cell system?

#5. Figure S1. "ORP8 knockdown caused significant mislocalization of both KRASG12V and LactC2 from the PM in each cell line tested, with an accumulation of LactC2 at the ER, and KRASG12V at the perinuclear region, as visualized by confocal microscopy (Figure S1A)." I think that this statement would require double labelling with ER and other organelle (e.g. Golgi) markers.

#6. Figure 7. Please comment the high concentration (10⁻⁵ M) of c7 used to inhibit PI4KIIIalpha. Indeed, this compound has an affinity in the nanomolar range for PI4KIIIalpha (IC₅₀=6.3 nM, see Boura and Nencka, 2015).

Reviewer #3 (Comments to the Authors (Required)):

This manuscript by Kattan and colleagues proposes that KRAS4B, a small GTPase whose localization and clustering on the plasma membrane activates signaling pathways leading to cell proliferation, depends on phosphatidylserine (PS) in the plasma membrane (PM) for its localization and activity, and that this localization plays a significant role in KRAS-dependent oncogenesis. This

work builds on the authors' previously published experiments indicating KRAS binding specificity toward phosphatidylserine.

The authors used two independent methods to reduce levels of PS at the plasma membrane while monitoring the localization of a constitutively active mutant form of KRAS4B. In the first, they found that depletion/abrogation of ORP5 and/or ORP8, lipid transfer proteins that harness a concentration gradient in PI4P to transport PS to the PM, caused reductions in both PM-localized PS and KRAS, as well as reduced clustering activity, suggesting that PS is indeed required for KRAS PM localization. This reduction in PM PS was accompanied by concomitant increases in PM PI4P and PI(4,5)P₂, reflecting reduced depletion of PI4P from the PM by ORP5/8. KRAS activity, tracked by monitoring the activation of the MAPK pathway via phosphorylation of ERK, was reduced in CaCO-2 cells where Orp8 had been abrogated. Secondly, the authors depleted PM PI4P, the driving force behind PS transfer to the PM, by using a small molecule inhibitor, C7, to block its synthesis by PI4KA. Under these conditions, they again observed a reduction in PM localized PS and KRAS, as well as reduced clustering, further supporting a role for PS in KRAS localization.

As KRAS is a major target of anti-cancer therapies, the authors examined the effect of PM PS depletion (through shRNA depletion of ORP5, ORP8, or both) on the growth rates of several KRAS-dependent or KRAS-independent cancer cell lines, finding that ORP depletion significantly reduced both anchorage-dependent and anchorage-independent growth rates in KRAS-dependent cancer cell lines, but not KRAS-independent cell lines or lines carrying wild-type KRAS. Furthermore, treatment with PI4KA inhibitor potently reduced the viability of KRAS-dependent cell line. Analysis of cancer patient ORP expression and survival rates indicated that elevated ORP5 or ORP8 levels was associated with poor prognosis in pancreatic cancer patients, where KRAS mutations are frequently found, but had little or no effect on prognosis in lung or other types of cancer.

This manuscript convincingly establishes the requirement of PM phosphatidylserine for KRAS localization and activation at the plasma membrane, demonstrating similar effects via two independent methods of depleting PM PS. Furthermore, they show that targeting PM PS levels to interfere with KRAS activation has significant consequences for the growth of KRAS-dependent cancers. The work represents a significant contribution to our understanding of the mechanism of KRAS localization and activation, and may have therapeutic implications.

This work appears suitable for publication in Life Science Alliance. Nonetheless, I have the following concerns and suggestions:

In Figure 2A, the rightmost lane of the first (left) panel represents an immunoblot against Orp8 in a cell line (sgOrp8) where this gene has been abrogated using CRISPR/Cas9. Nonetheless, a band (though admittedly weaker) appears at the same size as Orp8 in the WT cells. It would be helpful if the authors could provide some explanation for this residual band. The overall conclusions from this figure, however, are not likely to be affected by this discrepancy.

The imaging data quantitated in figure 2B-E would be greatly augmented by the addition of some representative images of immunogold-labeled PM sheets used in the analysis. These could be included in a supplemental figure.

The representative confocal images shown in Figure 3B, 7A, and S1A are difficult to evaluate qualitatively without the representation of each protein/probe's localization individually, as the accompanying Manders coefficient analysis of each is based on the proteins/probes' co-localization with CAAX. As a consequence, the representative images provided do not add significant information to these figures. Representing each of the colored channels independently for the same cell view would be far more informative.

Authors erroneously indicate in the text and in Figure 1B that human Fig4 is a homolog to *C. elegans* Sac1. A BLAST search of *C. elegans* Sac1 against the human proteome yields a clear human ortholog, also called Sac1. References to Fig4, a different Sac domain-containing phosphoinositide phosphatase whose localization and phosphoinositide substrate specificity differ significantly from those of Sac1, should be removed from the manuscript.

Reviewer #1:

1. In Figure 2D the authors use a PtdIns(4)P reporter FAPPH (most likely FAPP1-PH-GFP) to quantitate the PM levels of PtdIns(4)P in ORP8-KO cells. FAPP1-PH in mammalian cells specifically recognizes the Golgi complex pool of PtdIns(4)P showing little if any localization to the PM (Godi et al. NCB 2004). More appropriate tools to evaluate the PM PtdIns(4)P levels are the P4M probe (Hammond et al JCB 2014) or the anti- PtdIns(4)P antibody used as described in Hammond Biochem. J. 2009.

The PI4P reporter is indeed FAPP1-PH-GFP.

To address the reviewer's concern, we obtained the probe and transfected CaCO-2 parental and ORP8 knockout cells. We imaged plasma membrane sheets using immuno-EM and quantified the PI4P content as gold-labelled GFP-P4M-SidM in new Figure 2E. The data shows a significant increase in PI4P levels on the plasma membrane in ORP8 knockout cells, indicative of reduced transport from the PM to the endoplasmic reticulum (ER). This is consistent with the data using the GFP-FAPP1-PH probe. We believe our original FAPP1-PH data is equally legitimate because we isolate plasma membrane sheets and so exclude the Golgi pool of PI4P, supporting previous work that FAPP1-PH binds to PM as well as Golgi PI4P (Balla et al. 2005). Thus, we directly visualize the consequence of reduced PI4P transport by ORP8 to the ER, with a concomitant reduced transport of PtdSer to the PM.

We have incorporated the new EM data in new Figure 2 and described the data on page 5 in the main text.

2. Again on Figure 2D, the methodology described by Hancock and Prior (2005) is very powerful in assessing the nanoclustering at the PM, nonetheless this represents a second level approach and it has to be coupled to more standard imaging techniques (i.e., confocal microscopy) in order for the authors to conclude that their manipulations induce significant changes in the localization of their probes. The authors follow this rule when addressing the effects of their manipulations on KRAS (KRASG12V-GFP) and PtdSer (LactC2-GFP) localization, they should do so also when it comes to assess the localization of FAPP1-PH-GFP or PLCd (again here most likely PLCd-PH-GFP).

The PIP₂ reporter is indeed PLCδ-PH-GFP.

We have edited the text (page 5) and now simply state that there is an increase in the amount of PI4P and PIP₂ on the plasma membrane rather than claiming a change in distribution. We agree with the reviewer that in regards to KRAS and PtdSer confocal imaging is imperative since we are talking about subcellular localization, however in the case of PI4P and PIP₂ we are simply assessing amount of each lipid on the plasma membrane which is difficult to quantify by confocal microscopy, but readily and accurately achieved by EM of intact PM sheets after immuno-gold labelling of the cognate probe. In the revised text, we now make statements and draw conclusions only about PM levels of PI4P and PIP₂ and not their subcellular distribution.

3. In a recent publication Venditti et al. showed that PtdSer is transported by the lipid transfer protein ORP10 to the trans Golgi complex from where most probably it gets transported to the PM (Venditti et. Al JCB 2019). What is the contribution of this additional PtdSer pool to the recruitment of KRAS to the PM (or to endo-membranes)?

In their paper, Venditti et al. show that ORP10 transfers PtdSer from its site of synthesis at the ER to the trans-Golgi (TGN) and that TGN PtdSer levels correspond to activity levels of phosphatidylserine synthase I in the ER. They also show that ORP10 knockdown results in a significant reduction of Golgi PtdSer levels but no change in PM PtdSer levels. These data suggest that the Golgi pool of PtdSer does not contribute substantially to PM PtdSer content, and hence to KRAS PM localization. That said, we still do observe PtdSer and KRAS on the plasma membranes in ORP5/ORP8 double knockdown MCF-7 cells as well as in C7-treated MDCK cells. As discussed in the main text, one potential mechanism to partially sustain PtdSer levels on the PM in double knockdown cells may be through increased vesicular transport of PtdSer through recycling endosomes, which are enriched with this phospholipid (Matsudaira et al., 2017). However, it is also possible that PtdSer accumulated in the ER due to ORP5/8 inactivation, may be shuttled at an increased rate to the TGN via ORP10 for transport to the PM. However, since the PM levels of KRAS and PtdSer after ORP5/8 depletion are still significantly lower than in

WT/parental cells, we would again conclude that the Golgi pool of PtdSer is not a major contributor of KRAS PM localization. These points and the Venditti paper are now discussed in the revised manuscript.

4. The authors hypothesize that the effects they observe on AKT phosphorylation (pSer-473) might depend on the increased metabolic flux on PM from PtdIns(4)P to PtdIns(4,5)P₂ to PtdIns(3,4,5)P₃. This prediction needs to be corroborated by data aimed at assessing the actual levels of PtdIns(3,4,5)P₃ at the PM (i.e., by the use of AKT-PH-GFP as a reporter). Also the use of the anti- AKTpSer-473 antibody here is a bit confusing as phosphorylation in this specific serine can occur as a result of a feedback loop mechanism. I would suggest to use the anti AKTpThr-308 as a more direct measure of AKT activation downstream of PtdIns(3,4,5)P₃ production.

We agree with the reviewer. To further validate that ORP8 knockdown leads to increased amounts of PIP₃ on the plasma membrane, we transfected CaCO-2 parental and ORP8 knockout cells with GFP-AKT-PH, imaged plasma membrane sheets using immuno-EM and calculated the amount of labeled PIP₃ (new Figure 2G). The data show a significant increase in PIP₃ levels on the plasma membrane of ORP8 knockout cells due to the increased metabolic flux from accumulated PI4P on the PM.

Western blot analysis of our panel of pancreatic cancer cell lines following knockdown of ORP5 or ORP8 revealed an increase in the levels of AKTpThr-308 in MiaPaCa-2 knockdown cells compared to parental and empty vector control cells, while no AKTpThr-308 signaling was detected in parental or knockdown MOH cells (new Figure 6). There was also no significant difference in AKTpThr-308 levels between parental and knockdown cells in the PANC-1 and BxPC-3 lines. These results, as discussed in the revised paper, may possibly reflect different signaling outputs from the different KRAS point mutations in the cell lines (MiaPaCa-2: G12C, MOH: G12R, PANC-1: G12D) as well as KRAS dependency.

5. Some of the tools used in this study are not described in a sufficient detail. The FAPPH and PLCd probes, for instance are not described and there is no clear reference in the paper to the literature where one can find a more systematic description.

We now include a full description of each probe used in the main text with references as requested (page 4).

Reviewer #2:

#1. Abstract: "Inhibiting KRAS function through regulating PM lipid PtdSer content may represent a viable strategy for KRAS-driven cancers". Isn't this final statement too optimistic, given the pleotropic effects of plasma membrane electrostatics on various events, including cytoskeleton attachment, clathrin mediated endocytosis, numerous signaling cascades? It is unlikely that KRAS is the only protein sensitive to PS content, notably in non-cancerous cells.

The reviewer makes some valid points in response to our claim. However, we do provide compelling evidence that reducing PM PtdSer levels is selectively toxic to KRAS transformed cells, both as a consequence of KD of ORP5 and ORP8 or pharmacological inhibition of PI4KIII α . Similarly, there was no organismal toxicity associated with blocking PtdSer ER to PM transport in *C. elegans*, whereas activated let-60 signaling was suppressed. So we would submit that there is cause for optimism. In part this may reflect the exquisite binding specificity of the KRAS membrane anchor for PtdSer, which is essential of PM targeting, rather than a dependence on non-specific electrostatic interactions. We have revised the discussion to better justify our assertions in the abstract.

#2. C elegans experiments (Figure 1B). The results are clear cut: knocking down obr2 and to a lesser extent obr4 strongly diminishes a phenotype (MUV) that is caused by the hyperactivity of K-RAS. However, the sentences accompanying these results are difficult to follow. As stated by the authors, there is no clear homolog or orthologs of ORP5 and ORP8 in C elegans. However, blast analysis suggest that three genes encode for proteins related to ORP5/8: obr2, obr3 and obr4. The authors say that inhibiting the expression of any of these genes affect the MUV phenotype; yet in Figure 1B they show only the results for obr2 and obr4. In addition, in a following sentence, they refer to ORP5 and ORP8, not to obr: "Knockdown of either ORP5 or ORP8 expression potentially suppressed the MUV phenotype, with around 90% of the

population displaying a single vulva, similar to treatment with the MEK inhibitor U0126 as well as to the positive controls riok-1 and hoe-1, previously described as potent suppressors of the MUV phenotype". Please clarify the corresponding paragraph.

We apologize for the lack of coherence in the description. Obr-3 knockdown in *C.elegans* reduced the multivulva phenotype by 40% ($p < 0.05$) while inhibition of obr-2 and obr-4 suppressed the MUV phenotype by 62% and 93% ($p < 0.0001$ for both), respectively. Therefore, in Figure 1B, we only presented our two top hits. Furthermore, obr-2 and obr-4 had similar %identities with *OSBPL5* and *OSBPL8* compared to obr-3 with the human genes. Since human *OSBPL5* and *OSBPL8* are also highly homologous, we predict that obr-2 and obr-4 more accurately represent *OSBPL5* and *OSBPL8*. We have clarified the text in the revised manuscript and added obr-3 iRNA result in new Figure 1B.

#3. The Western Blot shown in Fig 2A is not convincing.

We have repeated the western blot with fresh protein samples and provide a clearer image.

#4. Please better justify the transition between the underlying strategies used in Figure 2 and Figure 3. In Figure 2, the authors used CRISPR/Cas9 to knock out ORP8 in CaCO-2 colorectal cancer cells. In Figure 3, the authors use interference RNA to knock down ORP5, ORP8 or both in MCF-7. The sentence accompanying this transition is "To further visualize KRAS and PtdSer mislocalization, ORP5 and ORP8 were knocked down separately as well as simultaneously in MCF-7 breast cancer cells (Figure 3A)." Why changing the cell system?

The reason is simply choosing the optimal cell for each imaging approach. Both are human cell lines: CaCO-2 human colorectal cancer cells yield excellent PM sheets for electron microscopy experiments, but are suboptimal for confocal imaging since they do not form a well-organized monolayer. MCF-7 cells, form a well-organized confluent monolayer and so yield better confocal images than CaCO-2, which in turn facilitates quantification of changes in subcellular distributions of PtdSer (LactC2) and KRAS using Manders coefficient. We have edited the text to justify the use of different cell systems.

#5. Figure S1. "ORP8 knockdown caused significant mislocalization of both KRASG12V and LactC2 from the PM in each cell line tested, with an accumulation of LactC2 at the ER, and KRASG12V at the perinuclear region, as visualized by confocal microscopy (Figure S1A)." I think that this statement would require double labelling with ER and other organelle (e.g. Golgi) markers.

We agree, we have therefore edited the manuscript to simply state that "ORP8 knockdown caused significant mislocalization of both GFP-KRASG12V and GFP-LactC2 from the PM in each cell line tested, with the accumulation of both probes on endomembranes, as visualized by confocal microscopy (Figure S2)."

#6. Figure 7. Please comment the high concentration (10-5 M) of c7 used to inhibit PI4KIIIalpha. Indeed, this compound has an affinity in the nanomolar range for PI4KIIIalpha (IC50=6.3 nM, see Boura and Nencka, 2015).

We agree with the reviewer that low concentrations of C7 are sufficient for PI4KIIIalpha inhibition. The C7 compound featured in Boura and Nencka 2015 was synthesized by Waring et al. who calculated the IC50 value of C7 to be 6.3nM by measuring the amount of IP₁ accumulated in cells treated with the inhibitor, as PI4P is phosphorylated to PIP₂ which is hydrolyzed to IP₃ and subsequently IP₁. However, when trying to define a more proximal marker of inhibition, they showed that 30μM of C7 resulted in the reduction of cellular PIP, PIP₂ and PIP₃ levels (Waring et al., 2014). Hence, we used concentrations ranging from 10-30μM for the confocal imaging experiments, while the concentrations used for the proliferation experiment ranged from 1 nM to 30μM (shown with a Log nM scale in the figure). We would speculate that the differences in potency between the in vitro biochemistry and in cell assays most likely reflect poor cellular penetration of the C7 compound.

Reviewer #3:

In Figure 2A, the rightmost lane of the first (left) panel represents an immunoblot against Orp8 in a cell line (sgOrp8) where this gene has been abrogated using CRISPR/Cas9. Nonetheless, a band (though admittedly weaker) appears at the same size as Orp8 in the WT cells. It would be helpful if the authors could provide some explanation for this residual band. The overall conclusions from this figure, however, are not likely to be affected by this discrepancy.

We have repeated the western blot with fresh protein samples and obtained a clearer image. The original band may have been a non-specific band.

The imaging data quantitated in figure 2B-E would be greatly augmented by the addition of some representative images of immunogold-labeled PM sheets used in the analysis. These could be included in a supplemental figure.

We agree with the reviewer and thank them for their suggestion. We now include representative scanning EM (SEM) images of immunogold-labeled PM sheets in Supplementary Figure 1.

The representative confocal images shown in Figure 3B, 7A, and S1A are difficult to evaluate qualitatively without the representation of each protein/probe's localization individually, as the accompanying Manders coefficient analysis of each is based on the proteins/probes' co-localization with CAAX. As a consequence, the representative images provided do not add significant information to these figures. Representing each of the colored channels independently for the same cell view would be far more informative.

As requested we have now added separate and merged images of GFP and mCherry channels for each representative image in new Figure 3B, 7A and S1.

Authors erroneously indicate in the text and in Figure 1B that human Fig4 is a homolog to C. elegans Sac1. A BLAST search of C. elegans Sac1 against the human proteome yields a clear human ortholog, also called Sac1. References to Fig4, a different Sac domain-containing phosphoinositide phosphatase whose localization and phosphoinositide substrate specificity differ significantly from those of Sac1, should be removed from the manuscript.

We thank the reviewer for pointing out this error. The reference to Fig4 is erroneous. The gene being knockdown by iRNA in *C. elegans* is the ortholog of human SAC1. We have corrected the revised manuscript (Page 3, new Figure 1B).

August 16, 2019

RE: Life Science Alliance Manuscript #LSA-2019-00431-TR

Prof. John F Hancock
University of Texas Health Science Center at Houston
Integrative Biology and Pharmacology
6431 Fannin Street, MSB 4.098
Houston, Texas 77030

Dear Dr. Hancock,

Thank you for submitting your revised manuscript entitled "Targeting Plasma Membrane Phosphatidylserine Content to Inhibit Oncogenic KRAS Function". As you will see, reviewer #1 appreciates the changes introduced in revision and we would thus be happy to accept your manuscript for publication in Life Science Alliance, pending final revisions to meet our formatting guidelines:

- Please upload all figure files as individual files
- Please provide your manuscript text as a word docx file
- Please add callouts in the manuscript to figure 1A
- Please render the scale bar in FigS2 more visible
- The section "Summary of Supplemental Material" can get removed from the manuscript

A. FINAL FILES:

- An editable version of the final text (.DOC or .DOCX) is needed for copyediting (no PDFs).
- High-resolution figure, supplementary figure and video files uploaded as individual files: See our detailed guidelines for preparing your production-ready images, <http://www.life-science-alliance.org/authors>
- Summary blurb (enter in submission system): A short text summarizing in a single sentence the

study (max. 200 characters including spaces). This text is used in conjunction with the titles of papers, hence should be informative and complementary to the title. It should describe the context and significance of the findings for a general readership; it should be written in the present tense and refer to the work in the third person. Author names should not be mentioned.

B. MANUSCRIPT ORGANIZATION AND FORMATTING:

Sincerely,

Reviewer #1 (Comments to the Authors (Required)):

The authors have now addressed my concerns to satisfaction. I therefore recommend this manuscript for publication in LSA.

August 19, 2019

RE: Life Science Alliance Manuscript #LSA-2019-00431-TRR

Prof. John F Hancock
University of Texas Health Science Center at Houston
Integrative Biology and Pharmacology
6431 Fannin Street, MSB 4.098
Houston, Texas 77030

Dear Dr. Hancock,

Thank you for submitting your Research Article entitled "Targeting Plasma Membrane Phosphatidylserine Content to Inhibit Oncogenic KRAS Function". It is a pleasure to let you know that your manuscript is now accepted for publication in Life Science Alliance. Congratulations on this interesting work.

DISTRIBUTION OF MATERIALS:

Again, congratulations on a very nice paper. I hope you found the review process to be constructive and are pleased with how the manuscript was handled editorially. We look forward to future exciting submissions from your lab.

Sincerely,
